# Patient satisfaction of breast reconstructive surgery following mastectomy in Brunei

**Shazana Nor**[1,2☯], **Koo Guan Chan**[2], **Hanif Abdul Rahman** [1,3]*, **Khadizah H. Abdul-Mumin**[1,4☯]

1 Pengiran Anak Puteri Rashidah Sa'adatul Bolkiah Institute of Health Sciences, Universiti Brunei Darussalam, Gadong, Brunei Darussalam, 2 Department of Plastics Reconstructive Surgery, Ministry of Health, Bandar Seri Begawan, Brunei Darussalam, 3 School of Nursing, University of Michigan, Ann Arbor, MI, United States of America, 4 School of Nursing and Midwifery, La Trobe University, Melbourne, Australia

☯ These authors contributed equally to this work.
* hanif.rahman@ubd.edu.bn

**Data Availability Statement:** Data cannot be shared publicly because the data belong to the Ministry of Health of Brunei as imposed by the

## Abstract

### Objective

To evaluate the impact of Breast Reconstructive Surgery (BRS) on patients' satisfaction and quality of life following mastectomy for breast cancer.

### Methods

A multi-method design study comprising quantitative and qualitative research was conducted between October to December 2019. The quantitative component consisted of a cross-sectional study using the Breast-Q questionnaire and the qualitative component involved in-depth interviews with eligible patients (N = 16) who underwent BRS following mastectomy for breast cancer. Quantitative analysis was performed including Fisher's exact test and One-way Analysis of Variance where a p-value of <0.05 was regarded as statistically significant. Qualitative data was thematically analysed using Braun and Clarke's thematic analysis.

### Results

14 out of 16 eligible patients participated in the study. We observed the lowest-scale score was for 'satisfaction with nipples' (mean score 32.7), followed by 'physical well-being: abdomen' (mean score 69.5). Despite a median score of 70 for 'satisfaction with breasts', patients were satisfied with the overall outcome of BRS (median score 80.5). Thematic analysis yielded three themes i.e. "I feel beautiful again" that described patients' satisfaction with aesthetic outcome with autologous reconstruction; "Striving for normality" that indicated BRS established back a sense of normality and improve their self-confidence and lastly, "I was well taken care of" highlighted the importance of providing 'well-informed' care to ensure overall satisfaction of their BRS journey.

Medical and Health Research Ethics Committee. The collected data used to support the findings of this study may be released upon request to the Ministry of Health of Brunei. With regards to the requested information on data availability, please find the contact email as follows: (bre.unit@moh.gov.bn).

**Funding:** The author(s) received no specific funding for this work.

**Competing interests:** The authors have declared that no competing interests exist.

## Conclusion

The uptake of BRS remains low since its availability in 2012, despite an overall increase in breast cancer cases in Brunei annually. Patients who underwent BRS have shown an increase BREAST-Q scores in breast satisfaction, psychosocial and sexual well-being after breast cancer treatment. Delivering high-quality patient-centred services and providing adequate information can influence the level of satisfaction for overall outcome. BRS should be considered as an important healthcare priority in Brunei and routinely be offered in the management of breast cancer.

## Introduction

Breast cancer is one of the most common cancers worldwide [1]. It remains the leading cancer and leading cause of death among the female population in Brunei [2, 3]. Early detection and treatment of breast cancer is key to long-term survival. The gold standard treatment for most operable breast cancers is mastectomy and guidelines from National Institute of Clinical Excellence (NICE) recommend breast reconstruction options to be offered to women undergoing breast cancer surgery [4]. Other countries have reported over 30% of patients requiring mastectomy had undergone breast reconstruction [5–7]. Globally, the mean age of patients undergoing mastectomy and breast reconstruction is approximately 50 years [8]. Notably, the survival rate of cancer patients is also increasing [9]. Additionally, evidence showed that patients who underwent breast reconstruction were more satisfied compared to mastectomy alone [10–14]. Reconstruction does not only restore the physical breast figure, but it also improves patients' psychological health after cancer treatment [11, 14, 15]. It has been shown that women who undergo mastectomy without breast reconstruction report a loss of perceived femininity, depression, anxiety, marital and sexual dysfunction [16]. Fanakidou et al also reported patients without breast reconstruction had higher level of loneliness, which was found to be correlated with poor health-related quality of life and higher levels of anxiety compared to those who had undergone breast reconstruction [16]. Therefore, it is worth highlighting that breast reconstructive surgery (BRS) plays an important role in breast cancer management as it can improve the overall quality of life of patients diagnosed with breast cancer who had undergone mastectomy [9].

Patients who had delayed reconstruction experienced a period of amastia which can have detrimental effects on their psychological well-being. Immediate reconstruction has been associated with high quality of life and patient satisfaction, as well as a positive psychological impact [17, 18]. However, immediate reconstruction may not be indicated for everyone as those requiring adjuvant radiotherapy would affect the reconstructed breast outcome. Some patients may also need time to recover from their breast cancer diagnosis and treatment before considering BRS. The importance of adequate pre-operative information from the plastic surgeon i.e. indications, benefits and risks of BRS, can lead to higher satisfaction to overall outcome [19]. This ensures that patients are psychologically prepared for and aware of the expected outcomes, therefore able to make a well-informed decision prior to undertaking major reconstructive surgery.

BRS has only been available in Brunei since 2012. Before this, only external breast prostheses were offered and our patients found it uncomfortable, caused rashes and difficult to maintain its place in bras. Most importantly, our patients did not acknowledge the prosthesis as part of their own body and often regarded it as a reminder to the cancer they once had. This

**Table 1. Number of breast cancers diagnosed from pathological specimen in Brunei.**

| Year | 2012 | 2013 | 2014 | 2015 | 2016 | 2017 | 2018 |
|---|---|---|---|---|---|---|---|
| Breast Cancer Diagnosis | 79 | 78 | 99 | 118 | 139 | 139 | 132 |

was a common reason amongst our patients wearing external prostheses following mastectomy in regaining a semblance of femininity. Reported reasons of women opting for BRS in one study include getting rid of the external prostheses, to be able to wear many different type of clothing, to regain feminine appearance and to feel whole again [20]. The Plastic Reconstructive Surgery department in the main largest hospital in Brunei is currently the only place that provides breast reconstruction services, which mainly offers free muscle-sparing transverse rectus abdominis myocutaneous (TRAM) flap, free deep inferior epigastric perforator (DIEP) flap and implants. Autologous breast reconstruction is most preferred within the Bruneian population as it gives a better cosmetic result and provides a more natural bulk/volume to the unilateral breast mound. Only one patient has undergone breast implants (with tissue expanders) surgery as she presented with previous bilateral mastectomies. Other studies have also shown higher satisfaction in patients who have opted for autologous reconstruction [15, 21].

Despite the increasing incidence of breast cancer in Brunei from 2012 to 2018 (**Table 1**), only 23 patients have undergone some form of BRS following mastectomy within the same time frame. Therefore, to address this deficiency, we aim to assess the impact of BRS on patients' satisfaction and quality of life following mastectomy for breast cancer. Our study consisted of both quantitative (BREAST-Q questionnaire) and qualitative (in-depth individual interviews) components. The quantitative part of this study evaluates important outcomes associated with quality of life following BRS such as physical well-being, psychosocial well-being and sexual well-being. To provide further insights, the qualitative part of this study aimed to further explore the patients' in-depth account of their experiences of quality of life following mastectomy for breast cancer. Hence the research questions were twofold: "What are the experiences of patients following mastectomy for breast cancer and how these experiences influence their overall well-being and quality of life?"

## Materials and methods

### Study design

A multi-method design study comprising both quantitative and qualitative research was conducted between October to December 2019 at our department in the main largest hospital in Brunei. The quantitative component consisted of a cross-sectional study using the Breast-Q questionnaire and the qualitative component involved in-depth interviews with the eligible patients. Ethics approval was obtained from the Institute of Health Sciences Research Ethics Committee (IHSREC), Reference number: UBD/PAPRSBIHSREC/2019/11. Principles outlined in the Declaration of Helsinki for all human or animal experimental investigations have been followed. Breast-Q instrument was used with permission from Mapi Research Trust. STROBE checklist for reporting observational studies was used to guide reporting of quantitative section of this study while COREQ checklist for reporting qualitative studies was used to guide reporting of the qualitative section of this study [22].

### Participants selection and recruitment

Between 2012 (the start of BRS services in Brunei) and 2019 (end of our study), only a total of 23 patients have undergone some form of BRS in our department. 16 out of the 23 patients

were eligible to be included in this study as they had undergone free muscle-sparing TRAM flap, free DIEP flap and implant surgery following mastectomy for breast cancer. All our patients received non nipple-sparing mastectomy (decision made by their breast surgeons). The remaining 7 patients were excluded from the study due to these reasons: (1) other diagnoses that were not breast cancer (2) left the country (3) palliative toilet and wound coverage. Out of the 16 eligible patients, only 14 patients consented to complete the Breast-Q questionnaire and partake in the interview.

## Quantitative section

**Instrument.** For the quantitative part of this study, the BREAST-Q post-operative reconstruction module (version 1.0) was utilised to assess the effects of BRS on all domains of quality of life and patient satisfaction. BREAST-Q is jointly owned by Memorial Sloan-Kettering Cancer Center and the University of British Columbia. Each response scale score is converted through the Breast-Q scoring software i.e. Q-Score$^{©}$ (The Q-Scores Company, Manhasset, New York, USA), into an equivalent Rasch Transformed Score, ranging from 0–100. There is no overall or total BREAST-Q score for each independent scale. The higher the score number indicates higher satisfaction or better health-related quality of life outcome [23]. However, a study has suggested dichotomizing score responses for each scale into 'satisfied' and 'not satisfied', where the cut-off for 'satisfied' responses were scores in the 75th percentile [19].

**Data collection procedure.** All eligible patients who went for their routine follow-up visit to our department were briefed about the study by distributing participant information sheets for both questionnaire and interview. Written consent and permission (which also included usage of excerpts, clinical photos, audiotaping and publication) has been obtained from participating patients. All completed questionnaires were assigned a Patient Identification Number (PIN) at random to when they returned it to our department. No names were retrieved from participants to ensure confidentiality and privacy. A reminder phone-call was given to those who have not returned their questionnaire.

## Data analysis

Data from patients' responses were tabulated on a spreadsheet and converted into a BREAST-Q score. Descriptive statistics and sub-group analysis including Fisher's exact test and One-way ANOVA were applied. RStudio$^{©}$ (RStudio Inc, Boston, MA, USA) was used for statistical analysis. A p-value of $<0.05$ was regarded as statistically significant.

## Qualitative section

**Data collection.** For the qualitative part of this study, patients were given a choice to partake in an individual interview or focused group discussion, which can be held on the same sitting as their clinic appointment or another date. An interview guide (**Table 2**) centred around a template of mostly open-ended questions adapted from interview items used by Klassen et al was utilised [24]. The interview guide ensure that the researchers maintain consistency in the data collected through asking questions focusing on the aim of the study which was guided by the interview guide [25]. This also prevented the interview from side-tracking and generating unwanted data that could have made the data analysis difficult [26]. The interview guide also facilitated the research team to ensure completeness of the data collected. The researcher also listened attentively to patients and probed necessarily where their answers required clarification, which is also a means to ensure participants' validation [27]. Patients were encouraged to tell their experience in their own words. Due to sensitive nature of some of the questions, referral to psychologist for appropriate counselling was made available to patients. All interviews

**Table 2. Interview guide.**

| Interview Guide |
|---|
| *Today we are going to discuss your experience after having undergone breast reconstructive surgery following your mastectomy procedure for breast cancer.* |
| *If you can recall the period before having your reconstructive surgery, did you know what surgical options were available in Brunei after mastectomy? What were your perceptions regarding breast reconstructive surgery?* |
| *When you have made the decision to undergo reconstructive surgery, what was going through your head? Did you have any particular concerns or expectations?* |
| *Let us talk through your post-operative period. How did you feel after your surgery? Were there any symptoms that you would like to talk about?* |
| *Now that you have recovered from your surgery, how has it affected you in your normal daily activities?* |
| *In terms of aesthetic outcome, how satisfied are you with your new breast? Are there any concerns or dissatisfaction that you would like to discuss?* |
| *How has the outcome of reconstructive surgery changed you psychologically? Has it affected your confidence level? Do you suffer from any emotional distress?* |
| *How has the outcome of reconstructive surgery affected any of your relationships with friends or family?* |
| *How has the outcome of reconstructive surgery affected your sexual life?* |
| *Overall, how satisfied were you with the surgical care provided to you throughout your breast reconstructive surgery journey? Were there any particular complaints/concerns you would like to voice out to improve our quality of care in the clinical setting?* |
| *Finally, has your outcome met your expectations? Do you have any regrets? Would you recommend this procedure to other patients?* |

were audio-taped and transcribed verbatim by the primary researcher (SN) who is a doctor in our department. Conducting the interview by the same researcher have also further ensured consistency of the data collected. Identity of patients remained concealed with their PIN. The recorded interviews were repeatedly listened to familiarise the data where the transcriber (SN) would be able to identify similar and recurring patterns of words.

**Data analysis.** All audio recorded data were first transcribed verbatim by SN. To ensure that data were accurately transcribed, KHA checked all transcriptions against the audio data. Thematic data analysis laid in by Braun and Clarke (2021) was performed by two research members (SN, KHA) concurrently following transcriptions. Iterative reading of transcripts was performed. This was to ensure familiarisation with and immersion into the data and gain deep understanding of the patients' experiences [28]. Next coding was conducted line-by-line, beginning with open-coding which then led to focus-coding. The data was systematically code by identifying and labelling meaningful units of information. Open coding were conducted inductively where codes were directly identified and formed from the transcripts [29]. Focus-coding were employed deductively by further grouping codes that shared similar meanings and expressions together and those codes that had distinct differences were maintained separately [30]. The coding and categorization process was cross-checked and any discrepancies were discussed until a final agreement was reached through 'member checking' which promotes trustworthiness of the study [31]. Constant comparative method was used to enhance analysis by comparing accuracy of transcriptions with the audio recorded data, data in one transcript to another and also data within the same transcript [32]. This process ensure that coding of data from the audio recorded, in the same transcript and other transcripts was compared until all data was accounted for [33]. Preliminary themes were formed and a consensus was reached upon finalisation of key themes and its sub-themes. No major disagreement occurred during the process of data analysis.

## Results

14 out of 16 patients (overall response rate 87.5%) consented to participate in the study. **Table 3** displays the demographic and clinical characteristics of patients who completed the

Table 3. Characteristics of patients who completed the BREAST-Q (n = 14).

| Characteristics | Value (%) |
|---|---|
| No. | 14 |
| Age at completion of questionnaire, year | |
| Mean +/- SD | 46.0 +/- 8.1 |
| Range | 28–56 |
| Age at reconstruction, year | |
| Mean +/- SD | 42.3 +/- 7.2 |
| Range | 27–53 |
| Age at mastectomy, year | |
| Median (IQR) | 37.5 (9.3) |
| Range | 27–49 |
| Marital status | |
| Single | 8 (57.1) |
| Married | 6 (42.9) |
| Chemotherapy | 12 (85.7) |
| Radiotherapy | 5 (35.7) |
| Timing of reconstruction | |
| Immediate | 4 (28.6) |
| Delayed | 10 (71.4) |
| Type of breast reconstruction | |
| TRAM | 4 (28.6) |
| DIEP | 9 (64.3) |
| Implant | 1 (7.1) |
| Laterality of surgery | |
| Unilateral | 13 (92.9) |
| Bilateral | 1 (7.1) |

SD, standard deviation; IQR, interquartile range.

BREAST-Q. Mean age at time of BRS was 42.3 +/- 7.2. Only four patients (28.6%) underwent immediate reconstruction whereas the majority underwent delayed reconstruction. Twelve patients (85.7%) received chemotherapy and five patients (35.7%) received additional radiotherapy. Majority of the patients (92.9%) opted for autologous breast reconstruction (n = 4 received free muscle-sparing TRAM flap and n = 9 received free DIEP flap) whereas only one patient underwent implant breast surgery.

Table 4 shows the median and mean of BREAST-Q scores from responses given for each BREAST-Q scale. Patients were extremely satisfied with the service provided by the plastic surgeon, medical and office staff. The lowest scoring scale was 'satisfaction with nipples' with mean score of 32.7, followed by 'physical well-being: abdomen' with mean score of 69.5. Despite a median score of 70 for 'satisfaction with breasts', patients were satisfied with the overall outcome of BRS (median score 80.5).

Sub-group analysis revealed that there were statistically non-significant results. However, there were several notable observations in the responses. It was observed that patients with higher reconstruction age were generally more satisfied in terms of breast, outcome, psychosocial well-being, information and plastic surgeon but less satisfied with physical well-being: chest, medical staff, and office staff. It was also observed that patients who had immediate reconstruction surgery were generally satisfied in all scales except for breast and physical well-being: chest. This was similarly seen in patients who underwent autologous breast reconstruction.

**Table 4. Distribution of BREAST-Q scores for each scale (n = 14).**

| Breast-Q scale | Median (IQR) | Range |
|---|---|---|
| Satisfaction with breasts | 70 (16.3) | 16–100 |
| Satisfaction with outcome | 80.5 (21.5) | 0–100 |
| Psychosocial well-being | 84 (37.0) | 23–100 |
| Physical well-being: Chest | 75.5 (24.3) | 53–100 |
| Satisfaction with information | 74.5 (16.3) | 0–100 |
| Satisfaction with plastic surgeon | 88 (19.0) | 45–100 |
| Satisfaction with medical staff | 100 (14.3) | 51–100 |
| Satisfaction with office staff | 95.5 (15.0) | 58–100 |
|  | **Mean +/- SD** | **Range** |
| Sexual well-being (n = 9) | 72.2 +/- 26.8 | 43–100 |
| Physical well-being: Abdomen (n = 13) | 69.5 +/- 20.3 | 47–100 |
| Satisfaction with nipples (n = 6) | 32.7 +/- 41.3 | 0–100 |

IQR, interquartile range; SD, standard deviation

From the patients' interviews, their satisfaction experience was depicted as a journey and three themes emerged (**Table 5**). All patients opted for individual interviews rather than focused-group discussion and none required additional psychological support.

## Theme one: "I feel beautiful again"

This theme represents patients' satisfaction with overall breast outcome after BRS. Many patients were satisfied with the aesthetic outcome of breast volume gained, especially with autologous reconstruction.

> *"I feel beautiful! Seriously, so beautiful again. (Patient gets emotional). I'm able to go shopping, put on whatever clothes I want. . . I feel alive. My confidence level is up. I've always been a positive person. . . After the surgery, I felt great."*

> (PIN104, 55 years old, 2 years post BRS)

All patients also shared their post-operative experience. Pain was commonly mentioned amongst patients and is undoubtedly uncomfortable but tolerable in the early post-operative period. 13 out of 14 patients that underwent free muscle-sparing TRAM or DIEP commented that the abdominal site was more painful than the reconstructed breast.

**Table 5. Themes and sub-themes derived from data-analysis of interview transcripts.**

|  | Themes | Subthemes |
|---|---|---|
| 1) | "I feel beautiful again." (Outcome after BRS) | • Breast cosmesis satisfaction<br>• Realistic expectations<br>• Post-operative issues |
| 2) | "Striving for normality." (Quality of life following BRS) | • Psychological well-being<br>• Sexual life<br>• Physical activities |
| 3) | "I was well taken care of." (Process of care) | • Doctor-patient relationship<br>• Surgical care from staff<br>• Recommendation for BRS |

*"The abdomen was the most painful part. I felt tightness and stiffness at the abdomen opera-tion site. . . I felt less pain at the breast operation site. I was in pain for three days or more, I needed help to get up and walk around."*

(PIN107, 56 years old, 7 years post BRS)

### Theme two: "Striving to normality"

This theme explores the patient's quality of life following BRS, in relation to her psychological, sexual and physical well-being. Some patients who underwent delayed reconstruction reported to have mourned over the loss of their breast after mastectomy. The availability of BRS has allowed them to establish a sense of normality and improve their self-confidence.

*"For me, it's like having my life back. . .When it comes to making decisions on getting my life back, I don't have much concern about what needs to be done. Even with the surgery for my cancer, my mastectomy, that was already the worst thing that one can go through. So nothing else after that should be worse, I think of it (BRS) as something that could benefit me."*

(PIN102, 53 years old, 5 years post BRS)

*"(My) confidence level and self esteem is up. Physically and mentally I'm improving. I felt very good. After my mastectomy, before reconstruction surgery, it (the experience) was heartbreak-ing. I was down for two years."*

(PIN113, 46 years old, 7 years post BRS)

Some patients shared how BRS had impacted their sexual life and relationship with their partners. Many of them commented feeling more sexually attractive, in spite of physical imper-fection of the reconstructed breast. A few patients stated a decline in intimacy post BRS, which was mainly due to discomfort during intercourse and appearance-related concerns. This is highlighted in the following excerpts:

*"We are not scared anymore. As husband and wife, I feel really confident. At first (after mas-tectomy), it felt quite different, he wouldn't touch me in a certain way, as if he was hiding something. But now, not anymore. Now, I don't feel like I am losing something."*

(PIN102, 53 years old, 5 years post BRS)

*"Without the breast, of course you don't feel sexy. Now I have a reconstructed breast, it feels sexy again. Truthfully speaking with our sexual life now, I don't go fully unclothed. I wear a nice sexy bra and that's good enough. He doesn't see me naked anymore. I don't want him to see my scars. I want him to feel good too. Now I have a reconstructed breast, it feels sexy again."*

(PIN104, 55 years old, 2 years post BRS)

This sub-theme explores the impact of BRS on physical function and daily routine activities. One patient who worked as a fitness instructor reported a decrease in physical activities.

*"I am unable to do the kind of workouts that I was able to do before the surgery."*

(PIN109, 28 years old, 1 year post BRS)

Some patients required assistance in the early post-operative recovery period.

*"Yes, the first two weeks after surgery were very difficult for me, especially when moving around. . . My husband supported me 100% so he was doing all the lifting, bathing and everything."*

(PIN102, 53 years old, 5 years post BRS)

Despite these few setbacks, the remaining patients stated they were able to return to their normal daily activities once fully recovered from BRS.

*"Does not affect my normal daily activities. I never thought that I had major surgery before."*

(PIN101, 50 years old, 8 years post BRS)

### Theme three: "I was well taken care of"

This theme highlights the importance of providing satisfactory care throughout a patient's reconstructive journey. An important topic expressed by most of the patients was maintaining a good relationship with the plastic surgeon. Most patients emphasised that adequate information about BRS and being shown clinical pictures of cosmetic outcomes from previous patients was helpful in leading their decision towards surgery.

*"I wanted to know what will happen (to my breast) after surgery and the plastic surgeon explained everything. He showed me examples through pictures. At first it looked painful but he explained further thoroughly."*

(PIN101, 50 years old, 8 years post BRS)

Overall, patients were quite satisfied with the surgical care provided from our staff including doctors, nurses and office staff. Their satisfaction can be exemplified in the excerpts below:

*"The ward staff, even though they're busy, are good, caring and attentive. General staff care is good. The Plastics team is very compassionate and makes me feel comfortable."*

(PIN104, 55 years old, 2 years post BRS)

*"You all are very wonderful, marvellous, excellent. From the staff, doctors, ward nurses, operating theatre—I was well taken care of."*

(PIN113, 46 years old, 7 years post BRS)

Conclusively, most patients were satisfied with their decision for BRS. Despite physical appearance issues and surgical discomforts, the majority did not express any regrets and would recommend BRS to other women with similar indications.

*"I would encourage them because the confidence level you'll get after reconstructive surgery is much higher."*

(PIN101, 50 years old, 8 years post BRS)

*"Nothing that I regret. If possible, I would like to assure women after undergoing mastectomy to consider the surgery (BRS). Please do it (BRS). It is worth doing. It is worth the pain."*

(PIN114, 43 years old, 4 years post BRS)

## Discussion

Despite the efforts established in Brunei to improve early breast cancer diagnosis, the partake of breast reconstruction procedures in affected patients remains a handful. Since October 2019, Brunei has offered a breast screening programme (with mammogram) for all women aged 40 years old and above. Any patients with abnormalities found during screening would be referred to the breast surgeon and would undergo the triple assessment (i.e. consultation, radiological imaging & core biopsy) within two weeks. Prevention programmes in all Well-Women Clinics at Primary Health Centres are also offered [34]. This includes self-physical examination and examination by doctors which should be made aware to the public in view of the incidence of breast cancer detected in the younger population [35].

Although BRS is known to improve physical and psychological well-being, Brunei still has a low uptake on breast reconstruction rate (only 23 patients) despite an increasing incidence of breast cancer annually between 2012 to 2018 (**Table 1**). This may be due to the lack of knowledge of availability of services, seeking for alternative treatment or other personal reasons. There is also the perception and stigma that others may deem BRS as cosmetic surgery. Patients with breast cancer are usually managed in a multidisciplinary team consisting of the breast surgeons, oncologist, radiologists and specialist nurses here in Brunei. Once a decision of mastectomy is made, the breast surgeon will refer eligible patients to our department for consultation of BRS. The breast surgeon performs all the mastectomy with or without axillary node clearance, depending on their sentinel lymph node biopsy (SLNB) results. The SLNB also decides whether patients will undergo immediate or delayed BRS i.e. if SLNB is positive, delayed reconstruction is advised as these patients will require immediate radiotherapy post mastectomy, which can affect aesthetic outcome to the reconstructed breast. A multi-center study in the United Kingdom (n = 141) and Italy (n = 384) found that adjuvant radiotherapy disproportionately increased ipsilateral revisions versus contralateral balancing surgeries (p = 0.028) [36]. In our study, five patients underwent adjuvant radiotherapy and all of them had delayed BRS. Younger patients also seemed to be much keen for breast reconstruction as they are usually more concerned with their physical appearance and femininity. We commonly observed that older (and married) patients in Brunei perceived their breast appearance as a less important aspect of their quality of life and therefore categorised BRS as "unnecessary surgery" for themselves.

The 'satisfaction with outcome' domain scale is considered to be the most important index to measure a patient's overall sense of satisfaction and whether her expectation is met with respect to aesthetic outcome [21]. **Table 4** shows that 11 out of 14 patients were satisfied with their outcome, although not statistically significant. However, as all eligible study populations in Brunei were invited to this study, practical clinical significance can be justified here.

Microsurgical abdominal flap breast reconstruction (e.g. TRAM and DIEP) is associated with higher levels of satisfaction and quality of life than expander-implant breast reconstruction [21]. Aesthetically, the vascularized abdominal skin and adipose tissue resemble the natural breast (i.e. softness, shape and symmetry) than that of a silicone implant. Therefore, in Brunei autologous reconstruction is the preferred option for surgeons and patients. Irrespective of whether a patient has immediate or delayed BRS, they either undergo free muscle-sparing TRAM and DIEP flap reconstruction and this decision is made intra-operatively by the plastic surgeon depending on the anatomical variant and route of the flap's perforators. DIEP is aimed first as it spares the transverse rectus abdominis muscle but vigorous dissection of the deep inferior epigastric vessels can compromise the vascularity of the flap, therefore muscle-sparing TRAM would be indicated. Although silicone implants and expanders are not readily available in our department, it is still offered. Cosmetically, a unilateral breast reconstructed

with a silicone implant looks different to the normal contralateral breast. Only one patient in our study underwent silicone implant surgery as she presented with bilateral mastectomies, thereby producing a better cosmesis outcome. Possible complications that are associated with implants include implant infection, implant malposition, implant rupture/leak and capsular contracture. Our study reinforced a first large study which was conducted in 325 Italian population with 133 (41%) underwent DIEP and 192 (59%) prosthetic reconstructions. The comparative study on autologous surgical techniques with breast implants showed evidence that DIEP led to the highest satisfaction in all BREAST-Q scores [9].

Satisfaction with nipples had the lowest scored scale in our study. A study analysing predictive factors of satisfaction showed that patients with nipple-sparing mastectomy who have undergone DIEP flap reconstruction achieved the highest BREAST-Q scores [37]. However, all our patients received non nipple-sparing mastectomy, a prior decision made by the breast surgeon due to oncological reasons. Therefore, we offered patients nipple reconstruction (using C-V flap technique) [38] following BRS. Seven patients opted for nipple reconstruction and only 3 of them further requested for micro-pigmentation of the nipple-areola complex. The final nipple aesthetic outcome may not be as satisfactory to some patients. In addition to that, patients recruited in this study were seen at different time-lines post BRS. As a result of that, overall outcome may be affected as patients who are still awaiting additional procedures to improve asymmetrical deformity might not be truly satisfied with their current aesthetic result. Other common procedures provided include fat graft injection, liposuction and mastopexy.

A statistically significant increase in BREAST-Q scores in breast satisfaction, psychosocial well-being, and sexual well-being were found in patients that underwent free muscle-sparing TRAM and DIEP flap reconstruction [15]. DIEP flaps have also been found to provide higher results in terms of satisfaction, particularly when associated with a nipple-sparing mastectomy, achieving the highest BREAST-Q scores in an Italian study involving 133 DIEP flap and 192 implant-based reconstructions [8]. However, radiotherapy, chemotherapy, hormone therapy, age and BMI had no influence on patient satisfaction. Additionally, these breast aesthetic gains are associated with a potential deterioration in physical well-being of the abdominal donor site e.g. abdominal hernias and unsightly long scars. During the interview sessions, patients mostly agreed that the abdominal site was the most painful during the post-operative and recovery period. A low mean BREAST-Q score of 69.5 for "physical well-being: abdomen" is also seen in **Table 4**.

Interviews conducted in our study enabled us to further gauge the impact of breast reconstruction on quality of life and satisfaction, from the participant's own perspectives. Common reasons for undergoing BRS were to improve confidence, self-image and comfortability under clothes. Patients that underwent immediate reconstruction does not have to experience a period of amastia, which can be an advantage to those that had delayed reconstruction. Immediate reconstruction has been associated with high quality of life and patient satisfaction, as well as a positive psychological impact [17, 18]. Patients who had delayed reconstruction experienced a period of amastia which can have detrimental effects on their psychological well-being. Living with amastia or wearing an external prosthesis often reminded of them of their cancer diagnosis. Prior to delayed reconstruction, some patients have expressed intimacy problems in their marriage since mastectomy due to reduced sense of femininity. After BRS, most of them commented that they feel more attractive when unclothed and sexually confident in their intimate relationship. BRS has shown to help strengthen the affective and sexual relationships of couples [14]. Zhong et al observed a statistically significant and clinically meaningful increase in improvement in breast satisfaction, psychosocial and sexual well-being in the early postoperative period after autologous BRS [15]. Therefore, BRS is not only known to

restore the physical breast volume figure, but it also improves overall patients' psychological health after cancer treatment.

Although the internet has widely become a good source for patients to access general information on BRS, the majority of the patients found that experiences from actual patients and being shown clinical postoperative photos were very useful. It is important to maintain good patients' interaction with their plastic surgeon and receive comprehensive pre-operative information. This enables them to make a well-informed decision which can significantly influence their level of satisfaction for breasts and overall outcome after BRS [12, 19, 39].

## Strengths and limitations

Although we have recruited almost all our study population, the number of cases is still small and results may not be generalizable. This study only looked at BREAST-Q scores post BRS as we have only recruited patients that have already undergone mastectomy and BRS. For future prospective studies, we may include the BREAST-Q questionnaire for pre-operative module before they undergo BRS. This may quantitatively show an improvement in psychological health and quality of life pre- and post BRS. We need to consider the pre-existing psychosocial well-being of patients prior to BRS, in order to strengthen post-mastectomy outcomes. There are also reduced variability in terms of breast reconstruction procedures by which we include largely autologous options (DIEP, TRAM) with only one patient undertaken implant-surgery. Patients recruited in this study were seen at different time-lines post BRS. Therefore, overall satisfaction outcome may be affected as some patients are still awaiting additional procedures to improve final aesthetic outcome. Qualitative findings based on patients' experiences were in-depth, although may not be an accurate recall. Interviews were done by the primary researcher who is also a doctor in our department. Future studies may consider having a professional who is not a member of our department to undertake role as this would reduce the impact of bias during interviews e.g. patients may have withheld certain emotions or criticism in front of a medical staff. Further studies should be a prospective one in which patients complete the questionnaire and participate in the interview within the same time-frame, i.e. one year post reconstruction, thereby complementing to the accuracy of data.

## Conclusion

This study is the first to investigate satisfaction outcomes of patients undergoing breast reconstruction post-mastectomy for breast cancer in Brunei. The results could potentially be used as baseline information for future patients to have reference on satisfaction and experience of BRS. It will provide information that could aid them with any concerns or misconception they have regarding BRS. BRS clinically provides some restoration to breast cosmesis mastectomy defects and also improves overall sense of satisfaction in psychosocial well-being after cancer treatment. Therefore, patients undergoing mastectomy for breast cancer deserve to be given the option for BRS. An increase in awareness regarding BRS and availability of its services will allow patients to seek consultation at early stages in their treatment. BRS can play a vital role in the management of patients with breast cancer and should be considered as an important healthcare priority in Brunei.

## Acknowledgments

The authors would like to express their utmost gratitude to all participants and nurse managers for supporting this study. We would also like to acknowledge the assistance of support from Dr PU Telisinghe and Dr PMB Aliuddin (Pathology Department) for providing us with the

breast cancer statistics, as well as AN RJG Pena for contacting all the patients to attend our clinic and lastly, to all the patients who participated in this study.

## Author Contributions

**Conceptualization:** Shazana Nor, Koo Guan Chan, Khadizah H. Abdul-Mumin.

**Data curation:** Shazana Nor.

**Formal analysis:** Shazana Nor, Hanif Abdul Rahman.

**Investigation:** Shazana Nor.

**Methodology:** Shazana Nor, Hanif Abdul Rahman, Khadizah H. Abdul-Mumin.

**Project administration:** Koo Guan Chan.

**Software:** Shazana Nor, Hanif Abdul Rahman.

**Supervision:** Koo Guan Chan, Hanif Abdul Rahman, Khadizah H. Abdul-Mumin.

**Writing – original draft:** Shazana Nor.

**Writing – review & editing:** Shazana Nor, Koo Guan Chan, Hanif Abdul Rahman, Khadizah H. Abdul-Mumin.

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
