## [Decision Letter · Decision Letter 0]

12 Dec 2022

PONE-D-22-31204Patient Satisfaction of Breast Reconstructive Surgery following MastectomyPLOS ONE

Dear Dr. Rahman,

Thank you for submitting your manuscript to PLOS ONE. After careful consideration, we feel that it has merit but does not fully meet PLOS ONE’s publication criteria as it currently stands. Therefore, we invite you to submit a revised version of the manuscript that addresses the points raised during the review process.

We look forward to receiving your revised manuscript.

Kind regards,

Fabio Santanelli, di Pompeo d'Illasi, MD, PhD

Academic Editor

PLOS ONE

Reviewers' comments:

Reviewer's Responses to Questions

**Comments to the Author**

1. Is the manuscript technically sound, and do the data support the conclusions?

Reviewer #1: Partly

Reviewer #2: Partly

2. Has the statistical analysis been performed appropriately and rigorously? 

Reviewer #1: Yes

Reviewer #2: Yes

3. Have the authors made all data underlying the findings in their manuscript fully available?

Reviewer #1: Yes

Reviewer #2: No

4. Is the manuscript presented in an intelligible fashion and written in standard English?

Reviewer #1: Yes

Reviewer #2: Yes

5. Review Comments to the Author

Reviewer #1: I have read with great interest the manuscript. Below my comments to the authors:

1. The “abstract” needs to be enriched with more information regarding the methodology and the results to make it clearer to the reader.

2. In the discussion section more studies with similar countries from other nations can be implemented (Persischetti et al DOI: 10.1007/s00266-022-02776-z).

3. In the material and methods and results section the data need to be mentioned in a more clear way. For example period of study is missing, patient characteristics, type of mastectomy and type of reconstruction etc. Please implement these information

4. Table 1 and 2 are incomplete and should be resubmitted.

5. References must be unified and edited as per journal recommendations.

Reviewer #2: The study is very interesting as it sheds light on patient-reported outcomes of breast reconstruction in a geographic area where this aspect of research has not been reported upon in a considerable manner so far. I commend the authors for their efforts. However, there are some areas of concern requiring some clarifications, revisions or corrections:

• Title: After carefully analyzing the contents of your manuscript, I would advise changing the title to the following: “Patient Satisfaction of Autologous Breast Reconstructive Surgery following Mastectomy in Brunei”. This addresses the population of patients your catered for more accurately and highlights the strengths of your content.

• Abstract: Please note that your abstract is short (163 words) and does not address the number of patients recruited in the study, the period in which the study was conducted, the inclusion and exclusion criteria, etc. The maximum word limit according to the publication’s submission guidelines is of 300 words. Consider improving your abstract with the additional details and data that are relevant to the readership.

• Introduction: Consider implementing how the mean age of patients undergoing mastectomy and breast reconstruction is of approximately of 50 years (PMID: 36376583), while survival of cancer patients is increasing, which supports the necessity for improved quality of life through breast reconstruction efforts (PMID: 35229192).

• Materials and methods: Please implement the period of study (from 2012 until when?). Please describe inclusion and exclusion criteria more clearly. Did you exclude patients who underwent no reconstruction whatsoever or conserving breast surgery? Did you include all types of reconstructions?

• Results: There are several points that should be raised and addressed:

a) Please state more clearly that only 16 out of the 23 breast reconstruction patients were selected, and that only 14 out of 16 agreed to participate in the study.

b) Additionally, I have some difficulty understanding which kind of reconstructive patients were recruited in the study. In your abstract you state that free TRAM, DIEP and implant-based reconstruction patients were included, but Table 2 includes only 14 patients out of which 4 received a TRAM (free?) and 9 received a DIEP flap. The total is 13, which means that something is missing or requires correction. This must be clarified.

c) Were implant-based reconstructions not taken into consideration for this study? If not, it should be discarded from the abstract and should be implemented as an exclusion criteria.

d) Consider implementing what kind of mastectomy the patients received. If a skin-sparing or radical mastectomy was selected, was the reconstruction assessed while having completed the reconstructive process? i.e. reconstruction of the nipple-areola complex (PMID: 33216178).

• Discussion: I believe several aspects could be implemented to improve on its contents:

a) Regarding abdominally-based breast reconstruction, you should consider implementing how DIEP flaps have been found to provide higher results in terms of satisfaction, particularly when associated with a nipple-sparing mastectomy, achieving the highest BREAST-Q scores in another study (PMID: 34559281). However, radiotherapy, chemotherapy, hormone therapy, age and BMI had no influence on patient satisfaction.

b) In your study, 5 patients underwent adjuvant radiotherapy. Did it affect the volume of the reconstructed breast volumes in a significant manner? (PMID: 34272176)

c) Consider discussing the efforts established in Brunei to improve early breast cancer diagnosis and to encourage breast reconstruction procedures in affected patients.

d) Regarding the limitations, you should implement the reduced variability in terms of breast reconstruction procedures which featured only autologous options (DIEP, TRAM).

• Consider implementing your references using a number instead of inserting the last name and year within the text. This makes tracking the references easier for the reader. Additionally, make sure you are using the Vancouver format for all your references (i.e. “Charmaz, K. (2014) Constructing grounded theory. sage.”, p. 17, Line 462).

• Table 1 appears to be partially unreadable. Make sure you correct the formatting before resubmitting it.

• Table 2 appears to be incomplete: while 4 patients underwent immediate reconstruction, you did not mention how many underwent delayed reconstruction. Additionally, were all immediate reconstructions free TRAMs? Were all the delayed reconstructions DIEP flaps? This could have caused a selection bias affecting patient-reported outcomes, and should also be mentioned in the study as a potential limitations.

• As a minor observation, while the overall contents of your manuscript are clearly understandable, consider having your manuscript proof-read by a native English speaker to correct very small language mistakes and improve overall readability (i.e. “through providing adequate information” in Abstract, p.3 Line 85).

6. PLOS authors have the option to publish the peer review history of their article (what does this mean?). If published, this will include your full peer review and any attached files.

Reviewer #1: No

Reviewer #2: **Yes: **Guido Firmani

---

## [Author Response · Author response to Decision Letter 0]

3 Jan 2023

Reviewer #1: Author’s responses

I have read with great interest the manuscript. Below my comments to the authors:

1. The “abstract” needs to be enriched with more information regarding the methodology and the results to make it clearer to the reader.

2. In the discussion section more studies with similar countries from other nations can be implemented (Persischetti et al DOI: 10.1007/s00266-022-02776-z).

3. In the material and methods and results section the data need to be mentioned in a more clear way. For example period of study is missing, patient characteristics, type of mastectomy and type of reconstruction etc. Please implement these information

4. Table 1 and 2 are incomplete and should be resubmitted.

5. References must be unified and edited as per journal recommendations.

Authors responses:

1. The methodology and results are detailed as per your comments. (Page 4 and 5)

2. This information is now added as per your comments (Page 19)

3. The requested information are now added (Page 7, 10, 11 and 12)

4. Table 1 and 2 reinserted after reformatting (Page 8 and 11)

5. References are now formatted according to the journal. 

Reviewer #2: 

The study is very interesting as it sheds light on patient-reported outcomes of breast reconstruction in a geographic area where this aspect of research has not been reported upon in a considerable manner so far. I commend the authors for their efforts. However, there are some areas of concern requiring some clarifications, revisions or corrections:

• Title: After carefully analyzing the contents of your manuscript, I would advise changing the title to the following: “Patient Satisfaction of Autologous Breast Reconstructive Surgery following Mastectomy in Brunei”. This addresses the population of patients you catered for more accurately and highlights the strengths of your content.

• Abstract: Please note that your abstract is short (163 words) and does not address the number of patients recruited in the study, the period in which the study was conducted, the inclusion and exclusion criteria, etc. The maximum word limit according to the publication’s submission guidelines is of 300 words. Consider improving your abstract with the additional details and data that are relevant to the readership.

• Introduction: Consider implementing how the mean age of patients undergoing mastectomy and breast reconstruction is of approximately of 50 years (PMID: 36376583), while survival of cancer patients is increasing, which supports the necessity for improved quality of life through breast reconstruction efforts (PMID: 35229192).

• Materials and methods: Please implement the period of study (from 2012 until when?). Please describe inclusion and exclusion criteria more clearly. Did you exclude patients who underwent no reconstruction whatsoever or conserving breast surgery? Did you include all types of reconstructions?

• Results: There are several points that should be raised and addressed:

a) Please state more clearly that only 16 out of the 23 breast reconstruction patients were selected, and that only 14 out of 16 agreed to participate in the study.

b) Additionally, I have some difficulty understanding which kind of reconstructive patients were recruited in the study. In your abstract you state that free TRAM, DIEP and implant-based reconstruction patients were included, but Table 2 includes only 14 patients out of which 4 received a TRAM (free?) and 9 received a DIEP flap. The total is 13, which means that something is missing or requires correction. This must be clarified.

c) Were implant-based reconstructions not taken into consideration for this study? If not, it should be discarded from the abstract and should be implemented as an exclusion criteria.

d) Consider implementing what kind of mastectomy the patients received. If a skin-sparing or radical mastectomy was selected, was the reconstruction assessed while having completed the reconstructive process? i.e. reconstruction of the nipple-areola complex (PMID: 33216178).

• Discussion: I believe several aspects could be implemented to improve on its contents:

a) Regarding abdominally-based breast reconstruction, you should consider implementing how DIEP flaps have been found to provide higher results in terms of satisfaction, particularly when associated with a nipple-sparing mastectomy, achieving the highest BREAST-Q scores in another study (PMID: 34559281). However, radiotherapy, chemotherapy, hormone therapy, age and BMI had no influence on patient satisfaction.

b) In your study, 5 patients underwent adjuvant radiotherapy. Did it affect the volume of the reconstructed breast volumes in a significant manner? (PMID: 34272176)

c) Consider discussing the efforts established in Brunei to improve early breast cancer diagnosis and to encourage breast reconstruction procedures in affected patients.

d) Regarding the limitations, you should implement the reduced variability in terms of breast reconstruction procedures which featured only autologous options (DIEP, TRAM).

• Consider implementing your references using a number instead of inserting the last name and year within the text. This makes tracking the references easier for the reader. Additionally, make sure you are using the Vancouver format for all your references (i.e. “Charmaz, K. (2014) Constructing grounded theory. sage.”, p. 17, Line 462).

• Table 1 appears to be partially unreadable. Make sure you correct the formatting before resubmitting it.

• Table 2 appears to be incomplete: while 4 patients underwent immediate reconstruction, you did not mention how many underwent delayed reconstruction. Additionally, were all immediate reconstructions free TRAMs? Were all the delayed reconstructions DIEP flaps? This could have caused a selection bias affecting patient-reported outcomes, and should also be mentioned in the study as a potential limitations.

• As a minor observation, while the overall contents of your manuscript are clearly understandable, consider having your manuscript proof-read by a native English speaker to correct very small language mistakes and improve overall readability (i.e. “through providing adequate information” in Abstract, p.3 Line 85).

Authors' responses:

The title has been changed to “Patient Satisfaction of Breast Reconstructive Surgery following Mastectomy in Brunei”.

“Autologous” not used as we include breast implant surgery in our study too

This is now addressed and changes have been made accordingly. (Page 4 and 5 - Abstract)

This has now been addressed accordingly (Page 6)

This is now addressed. (Page 7 and 8)

a) This is now addressed. (Page 8)

b) This is now addressed. (See Table 2 Page 11)

c) This has now been addressed (Page 11 and 19).

d) This has been addressed (Page 17 and 18)

a) This has now been addressed (Page 18 and 19)

b) This is now added in discussion. (Page 17)

c) This is now added in discussion. (Page 16 and 17)

d) This is now added. (Page 20)

This is now addressed.

Table 1 and 2 reinserted after reformatting (Page 8 and 11)

This is now added in discussion. (Page 18 and 19)

The paper was checked by native English speaker before resubmission.

---

## [Decision Letter · Decision Letter 1]

22 Feb 2023

PONE-D-22-31204R1Patient Satisfaction of Breast Reconstructive Surgery following Mastectomy in BruneiPLOS ONE

Dear Dr. Hanif Abdul Rahman,

Thank you for submitting your manuscript to PLOS ONE. After careful consideration, we feel that it has merit but does not fully meet PLOS ONE’s publication criteria as it currently stands. Therefore, we invite you to submit a revised version of the manuscript that addresses the points raised during the review process.

We look forward to receiving your revised manuscript.

Kind regards,

Fabio Santanelli, di Pompeo d'Illasi, MD, PhD

Academic Editor

PLOS ONE

Reviewers' comments:

Reviewer's Responses to Questions

**Comments to the Author**

1. If the authors have adequately addressed your comments raised in a previous round of review and you feel that this manuscript is now acceptable for publication, you may indicate that here to bypass the “Comments to the Author” section, enter your conflict of interest statement in the “Confidential to Editor” section, and submit your "Accept" recommendation.

Reviewer #1: (No Response)

Reviewer #2: All comments have been addressed

Reviewer #3: (No Response)

2. Is the manuscript technically sound, and do the data support the conclusions?

Reviewer #1: No

Reviewer #2: Yes

Reviewer #3: Partly

3. Has the statistical analysis been performed appropriately and rigorously? 

Reviewer #1: I Don't Know

Reviewer #2: Yes

Reviewer #3: Yes

4. Have the authors made all data underlying the findings in their manuscript fully available?

Reviewer #1: Yes

Reviewer #2: Yes

Reviewer #3: Yes

5. Is the manuscript presented in an intelligible fashion and written in standard English?

Reviewer #1: Yes

Reviewer #2: Yes

Reviewer #3: Yes

6. Review Comments to the Author

Reviewer #1: I would like to thank the authors for presenting a revised version of the manuscript. Nevertheless,

It is impossible for me to review the manuscript since the presented file doesn’t indicated the differences between the current and the previous version (i.e no track changes used for the word file, or red letters for the updated text. Moreover, the reply to the authors file presents only comments such as “this has now been addressed”. The authors should elaborate more on how they addressed the reviewers’ comments and add in their replies the modified part of the manuscript.

Reviewer #2: This is a revision to a previously assessed manuscript. The changes which have been made to the title, abstract, manuscript and tables are satisfactory. Well done. Some minor corrections are however still warranted:

• The abstract has been significantly improved upon, but now exceed the maximum word limit of 300 (it now counts 333 words): please trim it down a little.

• I advise against using “Plastic Reconstructive Surgery” as an acronym. It has been repeated 6 times throughout the manuscript. It is best to spell it out in full or to use “our department” instead.

• You still cited Cano et al. (2013) (Materials and methods, p. 8) in the previous reference style. Please use Vancouver style and number the reference.

• Please add the name of the company, the city and country of its headquarters next to the name of software and other copyrighted material. For example: Q-Score© (The Q Scores Company, Manhasset, New York, USA); RStudio© (RStudio Inc, Boston, MA, USA).

• Type of mastectomy has been successfully implemented in the discussion, however it needs to be mentioned in the Materials and Methods section as well.

• Since all patients received a non-nipple sparing mastectomy, you mentioned that some opted to receive nipple reconstruction while others did not. Has this data been recorded? Additionally, which techniques did you use? Consider referencing this paper by Paolini et al. for added background and a comprehensive description of techniques for nipple-areola complex reconstructions. (PMID: 33216178).

• Regarding abdominally-based breast reconstruction, please implement findings from this study from Santanelli di Pompeo et al. according to which patients were more satisfied with breast reconstruction when they received a DIEP flap compared to implants, particularly when associated with a nipple-sparing mastectomy, as demonstrated by higher BREAST-Q scores (PMID: 34559281). This is worthwhile mentioning in your discussion because no patient could receive a nipple-sparing mastectomy in your study for oncological reasons.

For the next round of revision, consider highlighting the corrected, added and modified elements from the manuscript using a different color such as a red font.

Reviewer #3: Thanks for the opportunity to review this study. This is study on an interesting subject for Brunei context. I am glad that the satisfaction and quality of life of women that undergo a breast reconstructive surgery following mastectomy was assessed. Important points are made. However, there are some points I would like to raise mainly around writing of the manuscript. See below my suggestions.

Abstract:

1. The abstract is very long and scattered. The authors should try to summarise the results of their study more concisely, considering the 300 words.

Introduction:

2. The part concerning the quality of life of these patients should be explored in more detail, highlighting which aspects are most affected and what is the experience of women undergoing BRS both before and after surgery. The authors could include further literature investigating this issue.

Method:

3. It is not clear who conducted the individual interviews or led the groups? Was it a psychologist, a mental health professional, or a doctor? The authors should clarify this.

4. What kind of open-ended questions were asked to obtain the qualitative data? The authors should state the questions.

5. The topics touched upon in the interview may have had an important resonance on the emotionality of the patients. Were they guaranteed appropriate psychological support in case they needed it?

6. The part on the analysis of the thematic data and the protocol used should be further investigated. It would be interesting to understand how the authors arrived at the themes and sub-themes presented in the study.

Results:

7. The authors should specify how many patients did an individual interview and how many chose to be in a group. This could be an important variable that could have an influence on the answers.

Discussion, limits and conclusion:

8. The discussions appear to be very long, dispersive, and not very clear in order to generate linear thinking with the results found and provide more insights. The authors should improve and implement these aspects.

9. It would be interesting to explore further the discussion regarding the results that emerged between patients who underwent imminent reconstruction and those who underwent it at a later step.

10. Authors could discuss and elaborate on the clinical implications of their results.

11. This study is characterised by the assessment of the satisfaction of patients undergoing BRS and their quality of life only to the post-operative phase. For this reason, the authors cannot report any improvement in the quality of life and in the psychological health of the patients. There is no longitudinal part that can confirm this. The authors should correct their manuscript (abstract, discussion, conclusions) specifying that high quality of life scores was found nowadays and not improvements.

12. The authors should be careful to claim that there has been an improvement in psychological health. This only seems to emerge to some extent from the qualitative analyses. Psychological health was not investigated quantitatively and through specific outcomes such as depression, anxiety, or psychological symptoms in general. The authors should review this aspect and consider whether to include it as a limit.

13. The authors should include as a limitation the fact that the study participants underwent reconstruction surgery at different times, and some are also awaiting further procedures to improve the aesthetic outcome. This might have influenced the results regarding the assessment of quality of life and satisfaction.

7. PLOS authors have the option to publish the peer review history of their article (what does this mean?). If published, this will include your full peer review and any attached files.

Reviewer #1: No

Reviewer #2: **Yes: **Guido Firmani

Reviewer #3: No

---

## [Author Response · Author response to Decision Letter 1]

24 Mar 2023

This is a revision to a previously assessed manuscript. The changes which have been made to the title, abstract, manuscript and tables are satisfactory. Well done. Some minor corrections are however still warranted:

• The abstract has been significantly improved upon, but now exceed the maximum word limit of 300 (it now counts 333 words): please trim it down a little.

• I advise against using “Plastic Reconstructive Surgery” as an acronym. It has been repeated 6 times throughout the manuscript. It is best to spell it out in full or to use “our department” instead.

• You still cited Cano et al. (2013) (Materials and methods, p. 8) in the previous reference style. Please use Vancouver style and number the reference.

• Please add the name of the company, the city and country of its headquarters next to the name of software and other copyrighted material. For example: Q-Score© (The Q Scores Company, Manhasset, New York, USA); RStudio© (RStudio Inc, Boston, MA, USA).

• Type of mastectomy has been successfully implemented in the discussion, however it needs to be mentioned in the Materials and Methods section as well.

• Since all patients received a non-nipple sparing mastectomy, you mentioned that some opted to receive nipple reconstruction while others did not. Has this data been recorded? Additionally, which techniques did you use? Consider referencing this paper by Paolini et al. for added background and a comprehensive description of techniques for nipple-areola complex reconstructions. (PMID: 33216178).

• Regarding abdominally-based breast reconstruction, please implement findings from this study from Santanelli di Pompeo et al. according to which patients were more satisfied with breast reconstruction when they received a DIEP flap compared to implants, particularly when associated with a nipple-sparing mastectomy, as demonstrated by higher BREAST-Q scores (PMID: 34559281). This is worthwhile mentioning in your discussion because no patient could receive a nipple-sparing mastectomy in your study for oncological reasons.

- Author Response to Reviewer:

- Abstract word count has been reduced to 300 words.

- “PRS” changed to “our” department (page 5,6,7,14,16,20)

- Cano et al. (2013) added

- Added on page 7

- Non nipple-sparing mastectomy included in Materials & Methods on page 6.

- Number of patient opted for nipple recon added. 

- C-V flap technique used

- Paper on C-V flap added as reference. Jalini et al (p19)

- Paper (Di Pompeo et al) included and referenced (p19)

# Reviewer 3

Thanks for the opportunity to review this study. This is study on an interesting subject for Brunei context. I am glad that the satisfaction and quality of life of women that undergo a breast reconstructive surgery following mastectomy was assessed. Important points are made. However, there are some points I would like to raise mainly around writing of the manuscript. See below my suggestions.

Abstract:

1. The abstract is very long and scattered. The authors should try to summarise the results of their study more concisely, considering the 300 words.

Introduction:

2. The part concerning the quality of life of these patients should be explored in more detail, highlighting which aspects are most affected and what is the experience of women undergoing BRS both before and after surgery. The authors could include further literature investigating this issue.

Method:

3. It is not clear who conducted the individual interviews or led the groups? Was it a psychologist, a mental health professional, or a doctor? The authors should clarify this.

4. What kind of open-ended questions were asked to obtain the qualitative data? The authors should state the questions.

5. The topics touched upon in the interview may have had an important resonance on the emotionality of the patients. Were they guaranteed appropriate psychological support in case they needed it?

6. The part on the analysis of the thematic data and the protocol used should be further investigated. It would be interesting to understand how the authors arrived at the themes and sub-themes presented in the study.

Results:

7. The authors should specify how many patients did an individual interview and how many chose to be in a group. This could be an important variable that could have an influence on the answers.

Discussion, limits and conclusion:

8. The discussions appear to be very long, dispersive, and not very clear in order to generate linear thinking with the results found and provide more insights. The authors should improve and implement these aspects.

9. It would be interesting to explore further the discussion regarding the results that emerged between patients who underwent imminent reconstruction and those who underwent it at a later step.

10. Authors could discuss and elaborate on the clinical implications of their results.

11. This study is characterised by the assessment of the satisfaction of patients undergoing BRS and their quality of life only to the post-operative phase. For this reason, the authors cannot report any improvement in the quality of life and in the psychological health of the patients. There is no longitudinal part that can confirm this. The authors should correct their manuscript (abstract, discussion, conclusions) specifying that high quality of life scores was found nowadays and not improvements.

12. The authors should be careful to claim that there has been an improvement in psychological health. This only seems to emerge to some extent from the qualitative analyses. Psychological health was not investigated quantitatively and through specific outcomes such as depression, anxiety, or psychological symptoms in general. The authors should review this aspect and consider whether to include it as a limit.

13. The authors should include as a limitation the fact that the study participants underwent reconstruction surgery at different times, and some are also awaiting further procedures to improve the aesthetic outcome. This might have influenced the results regarding the assessment of quality of life and satisfaction.

- Author response to reviewer 3

- Abstract word count has been reduced to 300 words.

- Added into Introduction

- Reaby LL to added on page 5

- SN is the primary researcher and a doctor in PRS department

- Table 2 Interview guide table inserted (page 8)

- Psychological support added.

- Two researchers spent a large amount of time iteratively reading and understanding the transcripts before performing coding and formulation of themes and subthemes. Several rounds of member checking was conducted to ensure discrepancies are minimized. Constant comparative method was used after each interview transcript to enhance analysis. Final themes and subthemes were formed with consensus and no major disagreement among researchers. (p8)

- All opted for individual interviews (page 12)

- Discussion has been improved

- Added in Discussion

- Abstract, Discussion and Conclusion amended

- Included as limitation 

- This limitation has been included

Thank you for your kind review and valuable comments.

---

## [Decision Letter · Decision Letter 2]

24 Apr 2023

PONE-D-22-31204R2Patient Satisfaction of Breast Reconstructive Surgery following Mastectomy in BruneiPLOS ONE

Dear Dr. Hanif Abdul Rahman,

Thank you for submitting your manuscript to PLOS ONE. After careful consideration, we feel that it has merit but does not fully meet PLOS ONE’s publication criteria as it currently stands. Therefore, we invite you to submit a revised version of the manuscript that addresses the points raised during the review process.

We look forward to receiving your revised manuscript.

Kind regards,

Fabio Santanelli, di Pompeo d'Illasi, MD, PhD

Academic Editor

PLOS ONE

Journal Requirements:

Reviewers' comments:

Reviewer's Responses to Questions

**Comments to the Author**

1. If the authors have adequately addressed your comments raised in a previous round of review and you feel that this manuscript is now acceptable for publication, you may indicate that here to bypass the “Comments to the Author” section, enter your conflict of interest statement in the “Confidential to Editor” section, and submit your "Accept" recommendation.

Reviewer #2: All comments have been addressed

Reviewer #3: (No Response)

2. Is the manuscript technically sound, and do the data support the conclusions?

Reviewer #2: Partly

Reviewer #3: Partly

3. Has the statistical analysis been performed appropriately and rigorously? 

Reviewer #2: Yes

Reviewer #3: N/A

4. Have the authors made all data underlying the findings in their manuscript fully available?

Reviewer #2: Yes

Reviewer #3: Yes

5. Is the manuscript presented in an intelligible fashion and written in standard English?

Reviewer #2: Yes

Reviewer #3: Yes

6. Review Comments to the Author

Reviewer #2: This is a revision to a previously assessed manuscript. While I am satisfied with the changes to what has been implemented beforehand, I believe that some changes are still needed regarding the formal aspects of your qualitative study’s methodology:

• I find that the defined objectives and research questions of your qualitative study are not expressed clearly enough, and they should be stated more explicitly at the end of your manuscript’s introduction.

• Please make sure you implement the COREQ checklist, or other relevant checklists listed by the Equator Network, such as the SRQR, to ensure complete reporting (http://journals.plos.org/plosone/s/submission-guidelines#loc-qualitative-research). You may specify in your manuscript’s methods which checklist was used.

• While you described participants’ selection and recruitment, the description of the sampling strategy, including rationale for the recruitment method, participant inclusion/exclusion criteria and the number of participants recruited is still lacking and should be improved upon.

• Data collection procedures are not detailed enough. For instance, was the assignment of a PIN done as a randomization strategy to guarantee respondents’ privacy? How did you assess quality of data? Did you take into account consistency of collection and completeness?

• The implementation of Table 2 improves your manuscript greatly. However, your data analysis procedures must be described in further detail to enable replication.

• Finally, please identify and discussion any potential source of bias, and add any missing limitation related to your methodology at the end of your discussion.

Reviewer #3: Thank you to the authors for addressing most of the concerns expressed above. I have further points to bring to their attention below:

1.“These reasons are common amongst women wearing external prostheses following mastectomy in regaining a semblance of femininity” Is this statement made by the authors in reference to their patients or is it generalisable to all women undergoing reconstruction with implants? If it is in reference to all, the authors should include the citation.

2.As suggested above, it would be interesting to explore in the introduction the psychological aspects of these women's quality of life, both pre and post reconstructive surgery. Several studies in the literature could be used, such as that of Fanakidou et al. (doi: 10.1007/s11136-017-1735-x) or Howard-McNatt (doi: 10.2147/BCTT.S29142). The authors should further investigate these aspects.

3.In the introduction all the variables that the authors will later investigate with their study should be presented. For this reason, a part should be devoted to the sexual well-being of women as this aspect is then discussed in the results. The authors should add this part and revise the whole introduction making sure to present the variables of interest adequately and thoroughly.

4.The discussion should be deepened with respect to the results that emerged on psychological well-being, sexual well-being, and general quality of life. This should be done by comparing with other studies and highlighting the clinical implications.

7. PLOS authors have the option to publish the peer review history of their article (what does this mean?). If published, this will include your full peer review and any attached files.

Reviewer #2: **Yes: **Guido Firmani

Reviewer #3: No

---

## [Author Response · Author response to Decision Letter 2]

4 Jun 2023

Reviewer #2:

This is a revision to a previously assessed manuscript. While I am satisfied with the changes to what has been implemented beforehand, I believe that some changes are still needed regarding the formal aspects of your qualitative study’s methodology:

• I find that the defined objectives and research questions of your qualitative study are not expressed clearly enough, and they should be stated more explicitly at the end of your manuscript’s introduction.

Author response: Thank you for your kind review and valuable comments. Aim and research questions provided in page 6 at the end of last paragraph.

• Please make sure you implement the COREQ checklist, or other relevant checklists listed by the Equator Network, such as the SRQR, to ensure complete reporting (http://journals.plos.org/plosone/s/submission-guidelines#loc-qualitative-research). You may specify in your manuscript’s methods which checklist was used.

Author response: COREQ checklist was implemented (Added in Page 6)

While you described participants’ selection and recruitment, the description of the sampling strategy, including rationale for the recruitment method, participant inclusion/exclusion criteria and the number of participants recruited is still lacking and should be improved upon.

Author response: Amended on Page 6. There were only 23 patients in total that had undergone BRS since services started in 2012.

Data collection procedures are not detailed enough. For instance, was the assignment of a PIN done as a randomization strategy to guarantee respondents’ privacy? How did you assess quality of data? Did you take into account consistency of collection and completeness?

Author response: Page 7. PIN given to ensure confidentiality & privacy. Page 9. Added on quality of data, consistency and completeness of data collection. 

• The implementation of Table 2 improves your manuscript greatly. However, your data analysis procedures must be described in further detail to enable replication.

Author response: Page 10. Data analysis procedures described in details

Finally, please identify and discussion any potential source of bias, and add any missing limitation related to your methodology at the end of your discussion.

Author response: Amended on Page 20.

Reviewer #3

Thank you to the authors for addressing most of the concerns expressed above. I have further points to bring to their attention below:

1.“These reasons are common amongst women wearing external prostheses following mastectomy in regaining a semblance of femininity” Is this statement made by the authors in reference to their patients or is it generalisable to all women undergoing reconstruction with implants? If it is in reference to all, the authors should include the citation.

Author response: Thank you for your kind review and valuable comments. Edited to “our patients” on Page 5.

2.As suggested above, it would be interesting to explore in the introduction the psychological aspects of these women's quality of life, both pre and post reconstructive surgery. Several studies in the literature could be used, such as that of Fanakidou et al. (doi: 10.1007/s11136-017-1735-x) or Howard-McNatt (doi: 10.2147/BCTT.S29142). The authors should further investigate these aspects.

Author response: Added referenced Howard-McNatt, as well as, Fanakidou has now been added in Page 4.

In the introduction all the variables that the authors will later investigate with their study should be presented. For this reason, a part should be devoted to the sexual well-being of women as this aspect is then discussed in the results. The authors should add this part and revise the whole introduction making sure to present the variables of interest adequately and thoroughly.

Author response: There were no specific variables we intended to investigate individually. All outcomes were derived from the Breast-Q questionnaires and the interview guide from Klassen et al. Amendments added on Page 5. 

4. The discussion should be deepened with respect to the results that emerged on psychological well-being, sexual well-being, and general quality of life. This should be done by comparing with other studies and highlighting the clinical implications.

Author response: Amendments added on Page 19.

---

## [Decision Letter · Decision Letter 3]

31 Jul 2023

Patient Satisfaction of Breast Reconstructive Surgery following Mastectomy in Brunei

PONE-D-22-31204R3

Dear Dr.Rahman,

We’re pleased to inform you that your manuscript has been judged scientifically suitable for publication and will be formally accepted for publication once it meets all outstanding technical requirements.

Kind regards,

Shimpei Miyamoto

Academic Editor

PLOS ONE

Additional Editor Comments (optional):

Reviewers' comments:

Reviewer's Responses to Questions

**Comments to the Author**

1. If the authors have adequately addressed your comments raised in a previous round of review and you feel that this manuscript is now acceptable for publication, you may indicate that here to bypass the “Comments to the Author” section, enter your conflict of interest statement in the “Confidential to Editor” section, and submit your "Accept" recommendation.

Reviewer #2: All comments have been addressed

Reviewer #3: All comments have been addressed

2. Is the manuscript technically sound, and do the data support the conclusions?

Reviewer #2: Yes

Reviewer #3: (No Response)

3. Has the statistical analysis been performed appropriately and rigorously? 

Reviewer #2: Yes

Reviewer #3: (No Response)

4. Have the authors made all data underlying the findings in their manuscript fully available?

Reviewer #2: No

Reviewer #3: (No Response)

5. Is the manuscript presented in an intelligible fashion and written in standard English?

Reviewer #2: Yes

Reviewer #3: (No Response)

6. Review Comments to the Author

Reviewer #2: This is a third revision to a previously assessed manuscript. Changes are satisfactory. I am happy with the manuscript in this current state. I commend the authors for successfully improving their manuscript.

Reviewer #3: Thank the authors for addressing the concerns expressed above. There are no further requests of revision. The article appears to explore important aspects of the psychophysical health of women undergoing mastectomy surgery in Brunei and may provide research and clinical insights into the topic.

7. PLOS authors have the option to publish the peer review history of their article (what does this mean?). If published, this will include your full peer review and any attached files.

Reviewer #2: **Yes: **Guido Firmani

Reviewer #3: No

---

## [Editor Report · Acceptance letter]

14 Aug 2023

PONE-D-22-31204R3 

Patient Satisfaction of Breast Reconstructive Surgery following Mastectomy in Brunei 

Dear Dr. Rahman:

I'm pleased to inform you that your manuscript has been deemed suitable for publication in PLOS ONE. Congratulations! Your manuscript is now with our production department. 

Kind regards, 

on behalf of

Dr. Shimpei Miyamoto 

Academic Editor

PLOS ONE